# *Streptococcus australis* and *Ralstonia pickettii* as Major Microbiota in Mesotheliomas

**DOI:** 10.3390/jpm11040297

**Published:** 2021-04-14

**Authors:** Rumi Higuchi, Taichiro Goto, Yosuke Hirotsu, Sotaro Otake, Toshio Oyama, Kenji Amemiya, Hitoshi Mochizuki, Masao Omata

**Affiliations:** 1Lung Cancer and Respiratory Disease Center, Yamanashi Central Hospital, Yamanashi 400-8506, Japan; r-higuchi1504@ych.pref.yamanashi.jp (R.H.); ootake.sotaro.gx@mail.hosp.go.jp (S.O.); 2Genome Analysis Center, Yamanashi Central Hospital, Yamanashi 400-8506, Japan; hirotsu-bdyu@ych.pref.yamanashi.jp (Y.H.); amemiya-bdcd@ych.pref.yamanashi.jp (K.A.); h-mochiduki2a@ych.pref.yamanashi.jp (H.M.); m-omata0901@ych.pref.yamanashi.jp (M.O.); 3Department of Pathology, Yamanashi Central Hospital, Yamanashi 400-8506, Japan; t-oyama@ych.pref.yamanashi.jp; 4Department of Gastroenterology, The University of Tokyo Hospital, Tokyo 113-8655, Japan

**Keywords:** mesothelioma, microbiome, 16S RNA sequencing, species

## Abstract

The microbiota has been reported to be correlated with carcinogenesis and cancer progression. However, its involvement in the pathology of mesothelioma remains unknown. In this study, we aimed to identify mesothelioma-specific microbiota using resected or biopsied mesothelioma samples. Eight mesothelioma tissue samples were analyzed via polymerase chain reaction (PCR) amplification and 16S rRNA gene sequencing. The operational taxonomic units (OTUs) of the effective tags were analyzed in order to determine the taxon composition of each sample. For the three patients who underwent extra pleural pneumonectomy, normal peripheral lung tissues adjacent to the tumor were also included, and the same analysis was performed. In total, 61 OTUs were identified in the tumor and lung tissues, which were classified into 36 species. *Streptococcus australis* and *Ralstonia pickettii* were identified as abundant species in almost all tumor and lung samples. *Streptococcus australis* and *Ralstonia pickettii* were found to comprise mesothelioma-specific microbiota involved in tumor progression; thus, they could serve as targets for the prevention of mesothelioma.

## 1. Introduction

The field of microbiome research was primarily initiated to study gastrointestinal diseases, such as pseudomembranous enteritis and irritable bowel syndrome; however, as the human intestinal microbiota is involved in carcinogenesis and cancer progression, it has also begun to focus on this area [1,2,3]. Moreover, new sequencing technologies have revealed bacterial flora in the pancreatic, lung, and breast tissues, in addition to the intestinal tissue [4,5,6,7,8].

Described as “the worst type of malignancy”, mesothelioma is a disease associated with extremely poor treatment outcomes and a 5-year overall survival of 3.4% [9]. Epidemiologically, mesothelioma is strongly correlated with asbestos inhalation and, since its onset, is usually observed approximately 40 years after asbestos inhalation. As of 2020, it is being increasingly reported worldwide [10]. Therefore, there is an urgent need to clarify its pathophysiology and establish methods for preventing its onset, as well as to introduce new treatments [11,12].

Several recent studies have reported the relationship between microbiota and carcinogenesis in colorectal cancer, oral cancer, pancreatic cancer, and lung cancer [13,14,15,16,17]; however, the significance of the microbiota in mesothelioma remains to be elucidated. Unlike previously studied oral, gastrointestinal and respiratory cancers, mesothelioma occurs in the thoracic cavity, and undergoes no direct interaction with external areas. Since the tumor environment of mesothelioma is presumably sterile, and since it is a rare disease, microbiome research on the involvement of the microbiota in the pathophysiology of this disease is lacking.

Here, the 16S rRNA of the bacterial genome of resected or biopsied mesothelioma specimens was amplified via polymerase chain reaction (PCR), followed by 16S sequence analysis via next-generation sequencing, to determine the composition of the microbiota and identify mesothelioma-specific bacterial flora. Furthermore, a predictive model for the onset of disease was developed based on these results, and the possibility of the prevention/control of mesothelioma onset was discussed.

## 2. Methods

### 2.1. Patients and Sample Preparation

In this study, eight patients who underwent surgical resection for mesothelioma at our hospital between January 2016 and August 2020 were enrolled unbiasedly. Since antibiotics treatment might modify bacterial composition, patients who had taken antibiotics orally or intravenously before surgery were excluded from this study. Written informed consent for genetic research was obtained from all the enrolled patients in compliance with the protocols of the Institutional Review Board at our hospital. The resected specimens were classified and staged according to WHO histological guidelines and the TNM *(Tumor-Node-Metastasis)* staging system, respectively [18]. Sections of formalin-fixed and paraffin-embedded tissues were stained with hematoxylin–eosin, followed by microdissection with the ArcturusXT laser-capture microdissection system (Thermo Fisher Scientific, Waltham, MA, USA), as previously reported [19,20,21,22,23,24]. For patients who underwent extra pleural pneumonectomy (EPP) and surgical resection of the lung, normal lung tissues, just under the visceral pleura, were also microdissected and examined. Since there were three surgical patients, eight patients and 11 specimens were analyzed. The GeneRead DNA FFPE Kit (Qiagen, Hilden, Germany) was utilized following the manufacturer’s instructions, and the DNA quality was examined by the use of primers for ribonuclease P [25]. In the same manner, tumor DNA was extracted from the FFPE (formalin-fixed paraffin embedded) samples obtained from patients with thymoma, the other rare malignant neoplasm in the thorax, and used as a control (*n* = 19).

### 2.2. 16S rRNA Amplification and Targeted Sequencing

The 16S rDNA V4 region was amplified by PCR and sequencing as previously described with minor modifications [7]. FFPE DNA was amplified with the Platinum PCR SuperMix High Fidelity (Thermo Fisher Scientific, Waltham, MA, USA) with forward primer 5′-GTGYCAGCMGCCGCGGTAA-3′ (16S_rRNA_V4_515F) and reverse primer 5′-GGACTACNVGGGTWTCTAAT-3′ (16S_rRNA_V4_806R). PCR products were confirmed by agarose gel electrophoresis and purified with Agencourt AMPure XP reagents (Beckman Coulter, Brea, CA, USA). End repair and barcode adaptors were ligated with an Ion Plus Fragment Library Kit (Thermo Fisher Scientific, Waltham, MA, USA) in compliance with the manufacturer’s instructions, and libraries were constructed. The library concentration was determined with an Ion Library Quantitation Kit (Thermo Fisher Scientific, Waltham, MA, USA), and the same quantity of libraries was set for each sequence. Emulsion PCR and chip loading were performed on the Ion Chef with an Ion PGM Hi-Q View Chef Kit, and sequencing was performed on the Ion PGM Sequencer (Thermo Fisher Scientific, Waltham, MA, USA). The sequence data were transferred to the IonReporter local server with the IonReporterUploader plugin. Data were analyzed with the Metagenomics Research Application using a custom primer set. The analytical parameter was set as the default.

### 2.3. Data Analysis

The original raw tags were obtained by merging paired-end reads using FLASH (v1.2.7), then they were filtered to obtain clean tags via Qiime (Version 1.9.1). The operational taxonomic units (OTUs) of the effective tags were classified and PCR chimeras were removed via Usearch (Uparse v7.0.1001) to identify the taxa composition of each sample with 97% identity. To obtain taxonomic assignments from phylum to species, the presentative sequence of each OTU was classified by taxonomy via the RDP (Ribosomal Database Project) classifier, with reference to the Silva (SSU123) 16S rRNA database, with confidence estimates of 80%.

### 2.4. Statistics

Continuous variables are described as the mean ± SD. One-way ANOVA (analysis of variance) and the Tukey–Kramer multiple comparison test were utilized to identify significant differences among groups. Statistical significance was defined as *p*-values below 0.05 in the two-tailed analyses.

## 3. Results

### 3.1. Patient Characteristics

In total, we analyzed 11 resected specimens from eight patients with mesotheliomas who had undergone surgery at our institution between January 2014 and August 2020. The clinicopathologic characteristics of the patients, including age, sex, histology, stage, smoking status, and performance of chemotherapy or extrapleural pneumonectomy (EPP), are shown in Table 1. Among the eight patients, all were males, seven were smokers and one was a non-smoker. According to the histological classification, there were six epithelioid, one sarcomatoid, and one biphasic mesotheliomas (Table 1). The eight patients enrolled in this study were classified according to TNM stage: stage IA (*n* = 1), IB (*n* = 4) and II (*n* = 3). The patients’ ages ranged between 53 and 78 years (68.1 ± 8.5 years). Seven patients underwent chemotherapy, and three patients underwent EPP.

### 3.2. OTU Analyses

Via OTU analysis, 61 OTUs were detected in 11 samples. The predominant (> 1% average relative abundance) classifiable OTUs involved two species, *Streptococcus australis* (abundance: 32.2 ± 29.6%) and *Ralstonia pickettii* (abundance: 24.4 ± 21.1%) (Figure 1). Both *Streptococcus australis* and *Ralstonia pickettii* were detected in the tumor tissues of six patients and in the lung tissues of all three patients who underwent EPP (Figure 1, Appendix A), and both species were identified in all mesothelioma tissues (Figure 1).

### 3.3. Differences in Microbiota between Mesotheliomas and Thymomas

To identify mesothelioma-specific microbiota, we compared the microbiota between mesothelioma and thymoma samples (Figure 2). The thymoma specimens showed no specific distribution of microbiota, and *Streptococcus australis* and *Ralstonia pickettii* were not detected either, suggesting that these species are specific to mesothelioma.

## 4. Discussion

The microbiota has recently been identified in some cancer tissues, including pancreatic and lung cancers, and its significance is attracting attention [4,5,6,7,8]; however, microbiome research focusing on mesothelioma is lacking. In this study, microbiome analysis was performed using resected mesothelioma specimens, and *Streptococcus australis* and *Ralstonia pickettii* were identified in almost all mesothelioma patients, with high levels of bacterial composition and abundance. Peripheral normal lung tissues adjacent to the tumor were also analyzed in patients who underwent EPP. *Streptococcus australis* and *Ralstonia pickettii* were detected in abundance in both the tumor and the adjacent lung tissues. Mesothelioma specimens and thymoma tissues (control) were analyzed simultaneously via the same process at the Genome Analysis Center in our institution, which determined that the specimens were not contaminated with these two bacterial species during the analysis process. By contrast, neither *Streptococcus australis* nor *Ralstonia pickettii* were detected in lung cancer tissues in recently published reviews of the microbiota [4,26,27,28]. Furthermore, the involvement of these two genera in the carcinogenesis of any organs has not been investigated. Since these two genera were detected in almost all mesothelioma patients, *Streptococcus australis* and *Ralstonia pickettii* may represent differential microbiome-related mechanisms in mesothelioma development.

Basic research on the lung microbiome has revealed that certain symbiotic bacteria form numerous micropores on the surface layer (visceral pleura) of the lungs of healthy individuals [29]. These micropores are formed by the secretion of cholesterol-dependent cytolysin (CDC), and there are five main types of CDC: pneumolysin, streptolysin, intermedilysin, mitilysin, and lectinolysin [30]. CDC, a pore-forming toxin, binds to the cholesterol on the cell surface and then polymerizes on the cell membrane to form transmembrane pores [29,31]. Furthermore, *Streptococcus pneumoniae, Streptococcus pyogenes, Streptococcus intermedius* and *Streptococcus mitis* are the major cause of CDC [30].

The *Streptococcus australis* identified in this study was first isolated from the saliva of children in Sydney, Australia, in 1991 [32]. During microbiological analyses of the saliva of children, Willcox et al. isolated strains of streptococci that could grow in media containing high concentrations of NaCl or KCl (up to 500 mM) [32]. These strains were initially identified as *Streptococcus mitis*, but were subsequently determined to be a separate species, according to DNA–DNA hybridization and biochemical analysis (Willcox, 1996) [33]. Nevertheless, based on 16S rRNA sequences, *Streptococcus australis* was shown to be clustered in the group corresponding to the *Streptococcus mitis* [32,34]. Based on the above information, it is likely that *Streptococcus australis* is present in the lung, particularly in the peripheral lung adjacent to the visceral pleura, in patients who develop mesothelioma, and it produces CDC in order to form numerous micropores in the visceral pleura on the lung surface. Furthermore, the pathophysiological hypothesis that asbestos microfibers pass through these micropores and reach the parietal pleura should also be considered. This hypothesis is consistent with the observation that ultra-thin fibers (about 0.02 μm in diameter) were the only asbestos fibers detected in the parietal pleura and mesothelioma tissues, and that the diameter of the micropores formed in the visceral pleura was estimated to be 250 Å (0.025 μm) [30]. However, there are many instances wherein mesothelioma does not occur even after asbestos inhalation. Differences in the composition of bacterial flora may contribute to individual differences in the occurrence of mesothelioma. This is a new hypothesis concerning the pathogenesis of mesothelioma, and further detailed investigation is urgently needed.

On the other hand, *Ralstonia pickettii* is a Gram-negative, rod-shaped bacterium [35]. *Ralstonia pickettii*, a Betaproteobacteria species, is a common microorganism inhabiting various environments, such as soils, rivers, and lakes. It is an oligotrophic organism, making it capable of surviving in nutrient-poor environments. The ability to use diverse organic compounds and survive in these harsh conditions makes *R. pickettii* useful for bioremediation [36]. *Ralstonia pickettii* is an emerging pathogen in clinical settings [37]. *R. pickettii* has come to be severely pathogenic in immunocompromised or fragile patients. Several medical institutions have reported outbreaks—patients with Crohn’s disease and cystic fibrosis in particular were found to be infected with *R. picketti*. Among the 55 reported cases of *R. picketti*. infection, most were due to contaminated saline solutions and sterile drugs [38]. These solutions are supposed to be contaminated during the manufacturing procedure, because *R. pickettii* is theoretically able to pass through the 0.2 µm filters that are generally used to sterilize medicinal products.

There are many indigenous microorganisms in the epithelia of several human organs (oral and auricular cavities, respiratory organs, gastrointestinal tract, skin, and reproductive organs), which play various roles in the body and have symbiotic relationships [1,3]. Disturbances in the bacterial flora (dysbiosis) change the risk of disease onset. Moreover, intestinal bacterial flora are relevant to numerous diseases, such as allergies, cancer, multiple sclerosis, Parkinson’s disease, depression, inflammatory bowel disease, and rheumatism [26]. Furthermore, the onset of these diseases has been alleviated and prevented via aseptic and specific pathogen-free processing in pathophysiological mouse models of the aforementioned diseases, and disease onset may also be prevented by improving the bacterial flora in humans [39]. If one or several organisms are the cause of disease, they may be a potential therapeutic target. In clinical practice for gastric cancer, carcinogenesis can be prevented by eradicating *Helicobacter pylori*, which is currently the standard treatment for the prevention of disease onset in infected patients [40]. Therefore, since the bacterial flora involved in the onset of mesothelioma has been identified, it may be possible to prevent the onset of mesothelioma in future clinical applications by controlling these two species. In particular, asbestos inhalation is a known cause of mesothelioma, and the prevention of mesothelioma is particularly important in high-risk populations exposed to asbestos [10]. The establishment of a probiosis model, with antimicrobial or vaccine therapy targeting the two target species identified in this study, may serve as a treatment regime for the prevention of the onset of mesothelioma.

This study has some limitations. First, the number of patients was small, owing to the extreme rarity of the tumor type, and the patients enrolled in this study were only Japanese. Second, no blood samples were analyzed for microbiota containing the two species *Streptococcus australis* and *Ralstonia pickettii*. The greater abundance of these two species in the tumor tissue may be associated with the impaired immunity of the tumor microenvironment, which may help these bacteria to proliferate in the blood, and thus they may be clinically applicable as serum biomarkers for mesothelioma. Third, it is unclear from our observational design whether the identified bacterial profiles are causally associated with oncogenesis, or are merely reflective of pathological processes in the mesothelioma. In this context, a larger series will be required to analyze the microbiome landscape of mesotheliomas more extensively, and to more clearly interpret the relevance of clinical variables via comprehensive multivariate analysis. However, since the major objective of this exploratory analysis was to identify the mesothelioma-specific microbiota that could be useful for clinical development, the modest sample size can still offer much insight.

## 5. Conclusions

This is the first study to examine the microbiota involved in mesothelioma, revealing two mesothelioma-specific species, *Streptococcus australis* and *Ralstonia pickettii*. Further research is required to reveal how the two species coexist with mesothelioma and how they are involved in the mechanism of carcinogenesis. In addition, by establishing probiosis models that can control these species, “precision medicine” can be developed for the prevention of the onset of mesothelioma. The results of this study might have clinical applicability, such as in preventing the onset of mesothelioma by controlling and enhancing the symbiotic bacterial flora through antibiotic or vaccine therapy, or in establishing a regular screening system in patients presumed to be at high risk for developing mesothelioma.

## Figures and Tables

**Figure 1 jpm-11-00297-f001:**
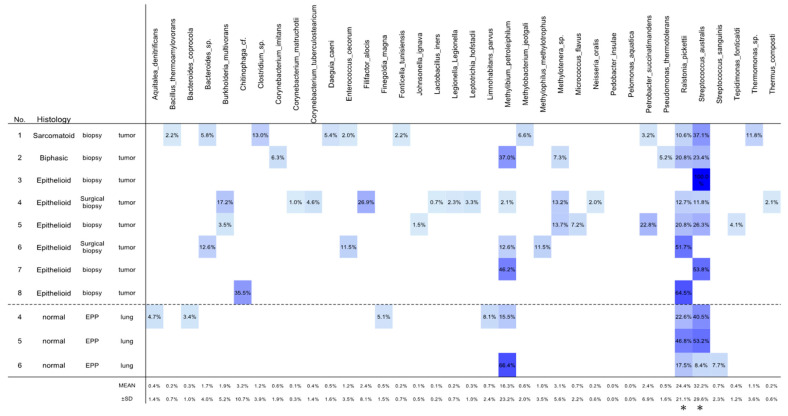
Composition and abundance of dominant species in all the samples. Heatmap visualizes the abundance of detected species. *, *p* < 0.05, compared with the other species except *Methylibium petroleiphilum*.

**Figure 2 jpm-11-00297-f002:**
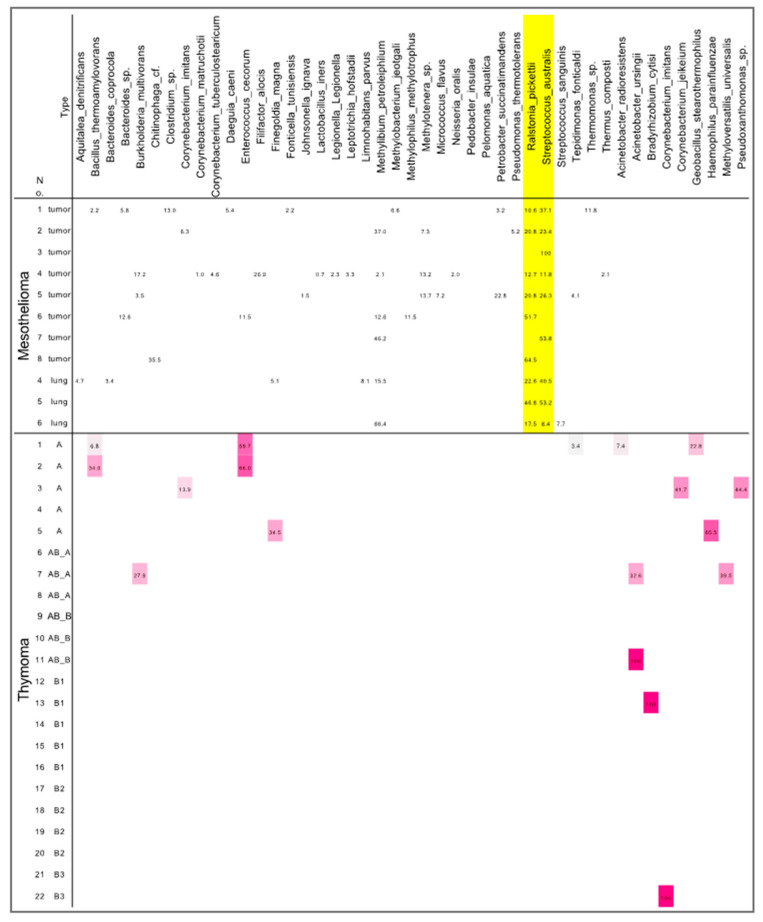
Microbiome differences between mesotheliomas and thymomas. *Streptococcus australis* and *Ralstonia pickettii* in the mesotheliomas are highlighted in yellow.

**Table 1 jpm-11-00297-t001:** Patient characteristics.

Parameter		Number of Patients	Overall Percentage
Total number		8	
Age (years), median (range)		71 (53–78)	
Sex			
	Male	8	100.0%
	Female	0	0.0%
Histology			
	Epithelioid mesothelioma	6	75.0%
	Sarcomatoid mesothelioma	1	12.5%
	Biphasic mesothelioma	1	12.5%
Stage			
	IA	1	12.5%
	IB	4	50.0%
	II	3	37.5%
Smoking Status (Pack year)			
	0	1	12.5%
	0 < PY ≦ 30	4	50.0%
	> 30	3	37.5%
Chemotherapy			
	Performed	7	87.5%
	Not performed	1	12.5%
EPP			
	Performed	3	37.5%
	Not performed	5	62.5%

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
