# Peer review of "Streptococcus australis and Ralstonia pickettii as Major Microbiota in Mesotheliomas"

_jpm, 2021, doi:10.3390/jpm11040297_

Round 1

Reviewer 1 Report

General comments :

Mesothelioma, a cancer linked to exposure to carcinogenic mineral fibers, is mainly caused by asbestos inhalation in the US and in the Western world. Genetic susceptibility may also contribute to the incidence of the cancer in certain families. So far, our knowledge about the tumor development is based on pathophysiological and toxicological researches and studies on tumor tissue samples, as well as cell lines from humans and experimental animals. However, the exact mechanism of the deadly disease has not been clarified. Based on results of modern research, further progress in mesothelioma development understanding and effective treatment discovery may be expected. Therefore, the subject of the manuscript is interesting. Although, there are some issues that should be addressed.

Specific comments:

There is a mistake in the title of the paper (i.e. methotheliomas) and several minor English errors in the text that need to be corrected (eg. „Emulsion PCR and chip loading was…” – should be were; „Analytical parameter was…” should be The analytical parameter was …” etc.

In the Introduction section of their manuscript, the Authors use the term: „irritable colitis”, although irritable bowel syndrome is the more accurate terminology for the disease.

I would recommend modifying the manuscript sections eg. Patient characteristics is included in the Results, so a reader may be confused.

I would prefer the manuscript should be arranged as follows:

  • Abstract
  • Introduction
  • Materials and methods
  • Results
  • Discussion
  • Conclusions
  • Acknowledgements
  • References

Conclusiom:

Minor revision is required.

Author Response

There is a mistake in the title of the paper (i.e. methotheliomas) and several minor English errors in the text that need to be corrected (eg. „Emulsion PCR and chip loading was…” – should be were; „Analytical parameter was…” should be The analytical parameter was …” etc.

Response: We corrected the misspelling of the title. We also had our manuscript checked and revised by English editing service of MDPI publisher.

In the Introduction section of their manuscript, the Authors use the term: “irritable colitis”, although irritable bowel syndrome is the more accurate terminology for the disease.

Response: I replaced the term with irritable bowel syndrome, as the reviewer kindly suggested.

I would recommend modifying the manuscript sections eg. Patient characteristics is included in the Results, so a reader may be confused.

Response: As a rule, the basement characteristics of patients are described in the Results section, not in the Methods section, in most of the research articles. I appreciate it if the reviewer would accept the current layout in our manuscript.

I would prefer the manuscript should be arranged as follows:

  • Abstract
  • Introduction
  • Materials and methods
  • Results
  • Discussion
  • Conclusions
  • Acknowledgements
  • References

Response: I agree with the reviewer. At this revision, I put the Methods section in front of the Results section.

Thank you for your thoughtful comments.

Reviewer 2 Report

This manuscript demonstrates dominant microbial species in patient tissues with mesothelioma by using an advanced 16S rRNA sequencing. They identified two specific species, Streptococcus australis and Ralstonia pickettii that were not found in the previous studies with lung cancers. Since there are only few studies are present regarding lung microbiota, this paper would be a good reference for future researches in this field. The reviewer would appreciate if the author would answer few minor questions:

  1. Title: Is “Methotheliomas” misspelling?

  1. Figure should not be duplicated. Figure 2 includes exactly same heatmap as Figure 1, but the labels of normal samples are different: "normal lung" in Fig. 1 and "non-tumor" in Fig. 2 that are confusing.

-Why don’t you combine Figure 1 and 2, or make Figure 1 different type of graph, such as a stacked column or table?

  1. Are S. australis and R. pickettii able to be lung dysbiosis markers worldwide in addition to Japan?

Author Response

  1. Title: Is “Methotheliomas” misspelling?

Response: Thank you for your suggestion. We corrected the misspelling in the title.

  1. Figure should not be duplicated. Figure 2 includes exactly same heatmap as Figure 1, but the labels of normal samples are different: "normal lung" in Fig. 1 and "non-tumor" in Fig. 2 that are confusing.

-Why don’t you combine Figure 1 and 2, or make Figure 1 different type of graph, such as a stacked column or table?

Response: As for the labels, we used the term “lung” instead of “non-tumor” in Figure 2. We also modified Figure 2 in order not to duplicate the heat-map graph.

  1. Are  australisand R. pickettii able to be lung dysbiosis markers worldwide in addition to Japan?

Response: As a limitation of the study, we examined the microbiota of mesothelioma only in Japanese patients. Therefore, further studies are necessary to examine whether our findings could be applicable to the other populations or nationalities.

We added these descriptions to the Discussion section.

Thank you for your thoughtful comments.